# Facile Reversible Eu$^{2+}$/Eu$^{3+}$ Redox in Y$_2$SiO$_5$ via Spark Plasma Sintering: Dwell Time-Dependent Luminescence Tuning

Fernando Juárez-López [1], Merlina Angélica Navarro-Villanueva [1,2], Rubén Cuamatzi-Meléndez [1], Margarita García-Hernández [3,4,*], María José Soto-Miranda [5] and Angel de Jesús Morales-Ramírez [5,*]

1 Instituto Politécnico Nacional, CIITEC IPN, Cerrada de Cecati S/N. Col. Santa Catarina, Azcapotzalco, Ciudad de México C.P. 02250, Mexico; fjuarezl@ipn.mx (F.J.-L.); mnavarrov@ipn.mx (M.A.N.-V.); ruben_c_m@yahoo.com.mx (R.C.-M.)
2 Instituto Politécnico Nacional, UPIEM-IPN, UPALM S/N Col. Lindavista, Gustavo A. Madero, Ciudad de México C.P. 07738, Mexico
3 Instituto Politécnico Nacional, CECyT No. 16 "Hidalgo"-IPN, Carretera Pachuca-Actopan Kilómetro 1+500, Distrito de Educación, Salud, Ciencia, Tecnología e Innovación, San Agustín Tlaxiaca, Hidalgo C.P. 42162, Mexico
4 Instituto Politécnico Nacional, UPIIH-IPN, Carretera Pachuca-Actopan Kilómetro 1+500, Distrito de Educación, Salud, Ciencia, Tecnología e Innovación, San Agustín Tlaxiaca, Hidalgo C.P. 42162, Mexico
5 Instituto Politécnico Nacional, ESIQIE-IPN, UPALM S/N Col. Lindavista, Gustavo A. Madero, Ciudad de México C.P. 07738, Mexico; msotom2001@alumno.ipn.mx
* Correspondence: mgarciah@ipn.mx (M.G.-H.); amoralesra@ipn.mx (A.d.J.M.-R.); Tel.: +52-5557296000 (ext. 55127) (A.d.J.M.-R.)

## Abstract

The present study investigates the luminescent behaviour of sol–gel derived Y$_2$SiO$_5$ powders doped with Eu$^{3+}$ ions, subjected to spark plasma sintering. The sintering process induces the partial reduction of Eu$^{3+}$ to Eu$^{2+}$, and the phenomenon is strongly dependent on the holding time within the SPS chamber. The luminescent properties are tunable via the initial Eu concentration, holding time and excitation wavelength, resulting in a wide range of emission colours from red (Eu$^{3+}$) at 220 nm excitation to blue (Eu$^{2+}$) at 365 nm, and mixed colours at 257 nm. Moreover, the Eu$^{3+}$/Eu$^{2+}$ redox process is reversible. Overall, the results demonstrate that SPS conditions can be exploited to modulate the valence state of luminescent centres, which is reversible by oxidation under ambient conditions, enabling controlled modulation of the optical properties.

**Keywords:** Y$_2$SiO$_5$; Eu; spark plasma sintering (SPS); europium reduction; luminescent properties; tunable emission spectra

## 1. Introduction

The search for advanced luminescent materials has intensified owing to their critical role in high-technology applications, including solid-state lighting, lasers, medical imaging and sensors [1,2]. The precise tuning of emission properties by chemical composition or excitation wavelength control is essential for optimising phosphors in lighting devices, displays, optical sensing, and biomedical markers [3,4]. One strategy involves co-doping, where distinct emission bands from separate luminescent centres enable colour-tunable output [5,6]. Europium (Eu) stands out in this context, offering two functional luminescent oxidation states: Eu$^{2+}$, showing a broad blue-green emission from 4f$^6$5d$^1 \longrightarrow$ 4f$^7$ transitions; and Eu$^{3+}$, showing a sharp reddish emission via $^5$D$_0 \longrightarrow$ $^7$F$_j$ transitions [7]. Consequently, numerous studies have focused on engineering Eu$^{2+}$/Eu$^{3+}$ co-doped systems for the development of multi-colour phosphors [8–13].

However, $Eu^{2+}$ lacks stable natural compounds and must be synthesised through the reduction of $Eu^{3+}$ precursors, and producing a sufficient amount of $Eu^{2+}$ remains challenging [14]. Conventional annealing in a reducing atmosphere composed of $H_2/N_2$ often degrades the host matrix or requires multi-step processing [15–17], while electrochemical reduction faces scalability limitations [18]. Recently developed methods include photochemical reactions [19,20] and reduction in aqueous systems via radicals [21]. An alternative approach exploits oxygen vacancy formation during materials processing, which facilitates in situ reduction even under oxidising conditions [22–24].

In general, phosphor synthesis results in the production of ceramic powders which are subsequently sintered to enhance the luminescent properties by eliminating structural defects and porosity; these defects act as non-radiative centres [25] that diminish the optical properties. Additionally, the thermal effects generated during consolidation improve crystallinity and promote dissolution of the luminescent centres, thereby increasing emission intensity [26,27]. Sintering also reduces light dispersion and scattering, which must be mitigated for applications such as displays, lasers, and LEDs [28,29]. Finally, sintering enhances the material's thermal and chemical stability, preventing degradation of the optical properties under subsequent exposure to high temperatures or UV radiation.

Among the various sintering techniques, spark plasma sintering (SPS) offers unique advantages, namely rapid densification at lower temperatures, coupled with intrinsic reducing conditions afforded by carbon dies and localised Joule heating [30–32]. Crucially rapid heating promotes oxygen vacancy formation, enabling $Eu^{3+} \longrightarrow Eu^{2+}$ reduction, as observed in some aluminosilicates [14,33,34].

Finally, regarding the ceramic frameworks that are responsible for transferring energy to the luminescent centres, the general research focus has moved from simple binary oxides to more complex ceramic systems, such as silicates, which present strong chemical stability, low thermal expansion, high conductivity, and an optimal optical damage threshold [35]. Of these silicates, yttrium oxyorthosilicate ($Y_2SiO_5$) has garnered significant attention for developing novel rare-earth activators. The $Y_2SiO_5$ compound exhibits two primary monoclinic polymorphs, X1 and X2, which significantly differ in their atomic arrangement. The low-temperature X1 phase, with the $P2_1/c$ space group, features isolated $SiO_4$ tetrahedra and two distinct yttrium sites: one where the yttrium ion is 7-coordinated in a distorted mono-capped trigonal prism geometry and another where it is 6-coordinated in a distorted octahedron, creating a layered structure interconnected by these polyhedra. In contrast, the high-temperature X2 phase, belonging to the $C2/c$ space group, also contains isolated $SiO_4$ tetrahedra but possesses a higher symmetry where both crystallographically unique yttrium sites are 6-coordinated, forming distorted octahedra that create sheets linked together by the silicate groups, resulting in a centrosymmetric framework distinct from the non-centrosymmetric X1 structure [36–38]. For luminescence properties, the rare-earth ions substitute for $Y^{3+}$ ions at both crystallographic sites.

Herein, we demonstrate facile, reversible $Eu^{3+} \Leftrightarrow Eu^{2+}$ redox control in doped $Y_2SiO_5$ via SPS, which consolidates the phosphor while tuning the valence of Eu. $Y_2SiO_5$: $Eu^{3+}$ (X1 phase) was synthesised by sol–gel processing and then sintered under varied holding times (HTs; 6, 15, and 30 min). SPS simultaneously induced a phase transition (X1 $\longrightarrow$ X2) and progressively reduced $Eu^{3+}$ to $Eu^{2+}$, with longer HTs enhancing the reduction efficiency. Furthermore, we demonstrated that the process can be fully reversed by air annealing. This work establishes SPS as a single-step route for consolidating phosphors while controllably modulating activator valence, eliminating external reductants.

## 2. Results and Discussion

### 2.1. Structural Evolution

Figure 1a shows the X-ray diffraction spectra of the sol–gel powder samples annealed at 1000 °C, for undoped samples and those doped with 1.0 and 2.5 at.% of $Eu^{3+}$. All sol–gel powders predominantly exhibit the $Y_2SiO_5$-X1 structure, which is a monoclinic structure with the $P2_1/c$ space group, comprising chains of $SiO_4$ tetrahedra connected with $Y^{3+}$ ions in 7 and 9 coordination (JCPDS 036-1476). This is the low-temperature phase, which has been commonly reported for sol–gel powders with luminescent properties, mainly because it can be obtained below 1100 °C [39–43]. Figure 1b,c show the structural evolution of the sintered coupons as a function of HT for the samples with 1.0 and 2.5 at.%, processed at 1200 °C. The spectra for the sol–gel synthesised powders are also included for ease of comparison. Above 1200 °C, a phase transformation typically occurs from X1 to X2 [44,45] the latter being a monoclinic structure belonging to the C2/c space group with 6 and 7 coordination (JCPDS 052-1810). This transformation is observed in the samples sintered via SPS, as shown in Figure 1. In all SPS-sintered cases, the dominant phase is X2, with minor reflections corresponding to the X1 phase, as well as the hexagonal $Y_{4.67}(SiO_4)_3O$ phase (JCPDS 030-1457). These phases are expected according to the equilibrium diagram and result from the incomplete X1 to X2 transition [46]. Additionally, for both Eu concentrations, an increase in HT enhances the crystallinity of the system. This is attributed to the greater thermal input, promoting a more complete phase transformation, given that it is a first-order reconstructive (non-diffusive) phase transition. There is an abrupt change in the crystal structure (symmetry $P2_1/c \longrightarrow C2/c$) and in the coordination of $Y^{3+}$, indicating a non-gradual atomic rearrangement that is therefore favoured by thermal input (supplied by an increase in HT). In the proposed system, the Eu content does not appear to cause a significant change in the transformation because no major differences in the diffraction patterns are observed between the two samples (Figure 1b,c).

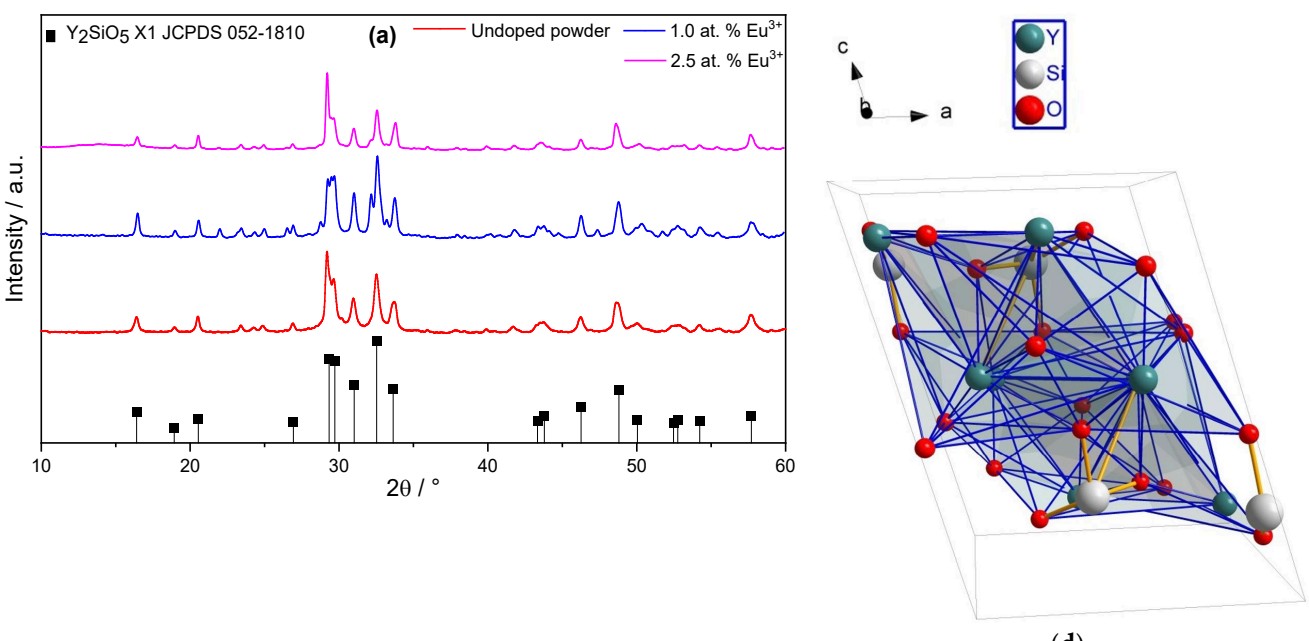

**Figure 1.** *Cont.*

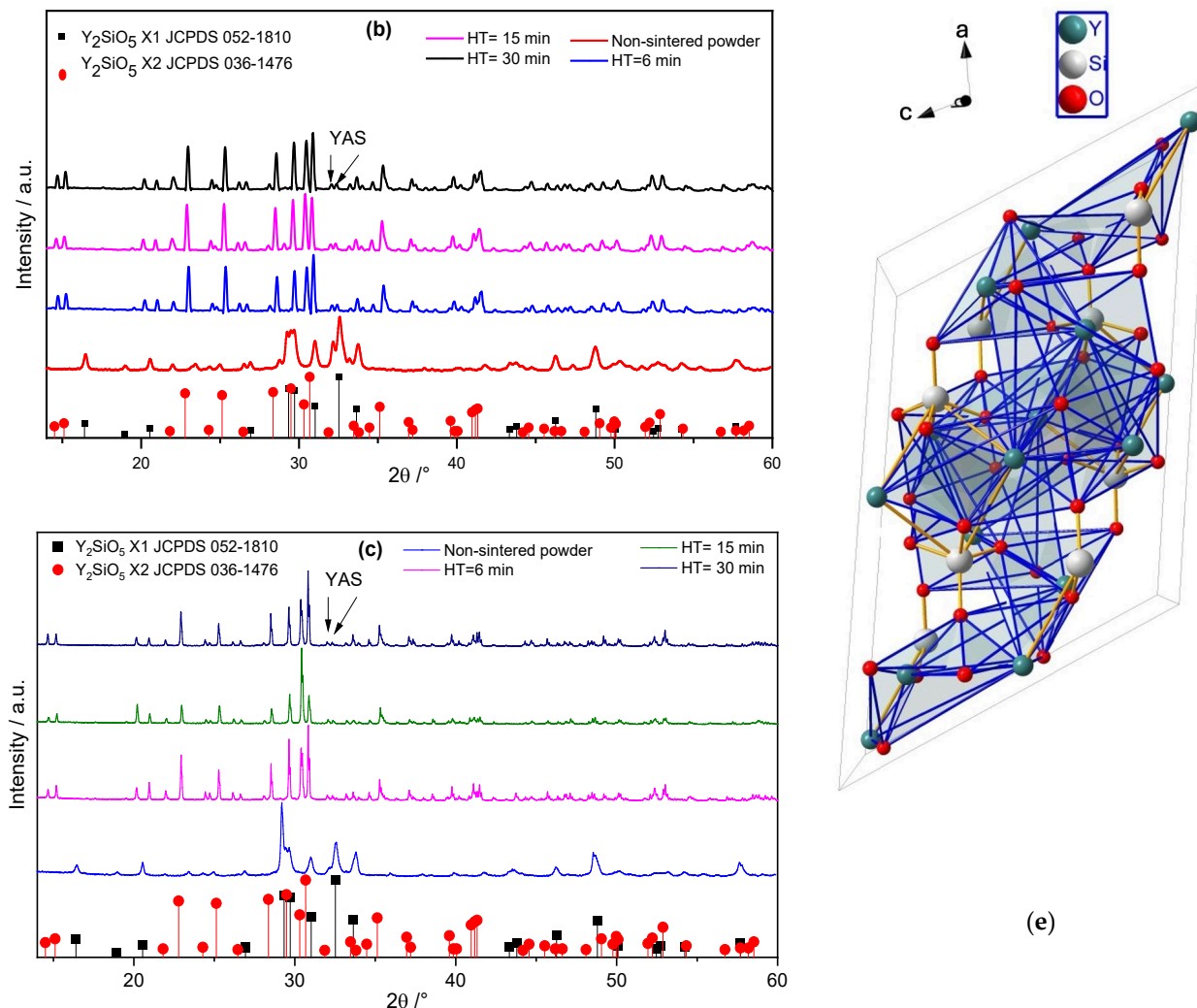

**Figure 1.** Structural evolution of $Y_2SiO_5$: $Eu^{3+}$. (**a**) Sol–gel derived powders, T = 1000 °C, (**b**) SPS coupons with 1.0 at.% Eu, as a function of HT, T = 1200 °C. (**c**) SPS coupons with 2.5 at.% Eu, as a function of HT, T = 1200 °C. (**d**) $Y_2SiO_5$-X1 structure. (**e**) $Y_2SiO_5$-X2 structure.

*2.2. SPS*

To verify the consolidation of the Eu-doped $Y_2SiO_5$ powders during the SPS process, the shrinkage of the samples was analysed during the process. Figure 2a shows a plot of the sintering rate (shrinkage for undoped and doped yttrium silicate powders as a function of the temperature), depicting the behaviour during SPS. The sintering rate is not affected by the Eu content; nevertheless, the sintering temperature can be varied in the range of 1100 to 1400 K for each powder. Moreover, after 1000 s of sintering at 1673 K, both undoped and doped Eu sintered compacts showed constant sintering rates, regardless of the Eu content, as observed in Figure 2b. The resulting sintered microstructure is expected to be similar in all coupons, and the Eu content did not alter the first sintering stage. Therefore, the sintering rate was merely due to volume diffusion, as is observed in Figure 2a, where the shrinkage rate as a function of the temperature is similar for all sintered coupons. However, a longer HT allowed for the complete reaction of Eu; in other words, a second sintering stage allowed for Eu reduction by enhancing vacancy diffusion during the SPS process, as suggested in Figure 2b, where the shrinkage rate did not change as a function of time.

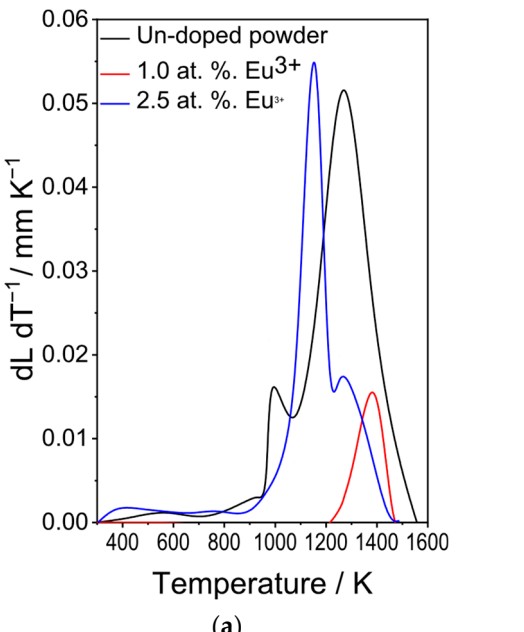 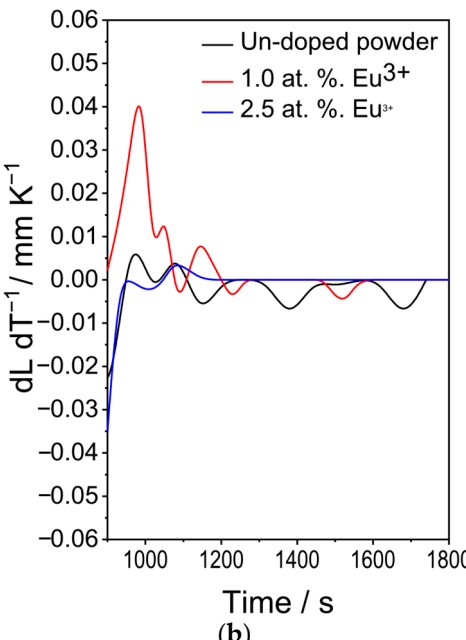

(**a**)             (**b**)

**Figure 2.** Plot of the sintering rate (shrinkage) for the undoped and doped yttrium silicate powder: (**a**) as a function of temperature, (**b**) as a function of time.

### 2.3. Luminescent Properties

#### 2.3.1. Sol–Gel Derived $Y_2SiO_5$: $Eu^{3+}$ Powders

Figure 3a shows the excitation spectrum ($\lambda_{em}$ = 616 nm) of $Y_2SiO_5$ powders doped with 2.5 at.% $Eu^{3+}$, displaying the two characteristic zones: the charge transfer (CT) band between 200 and 320 nm, and the f-f transition region between 350 and 450 nm. The first, centred at 257 nm, corresponds to the charge transfer $O^{2-} \longrightarrow Eu^{3+}$, whereas the second corresponds to the intramolecular $^7F \longrightarrow \, ^5D$ transitions of the $Eu^{3+}$ 4f electrons, specifically $^7F_0 \longrightarrow \, ^5D_4$ at 365 nm, $^7F_0 \longrightarrow \, ^5G_2$ at 381 nm, and $^7F_0 \longrightarrow \, ^5L_6$ at 393 nm [47].

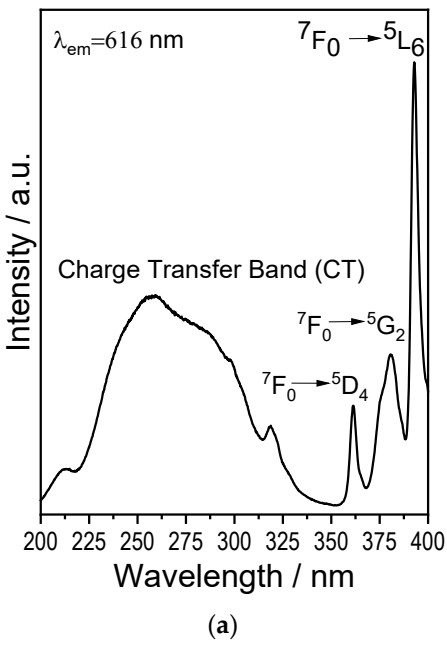 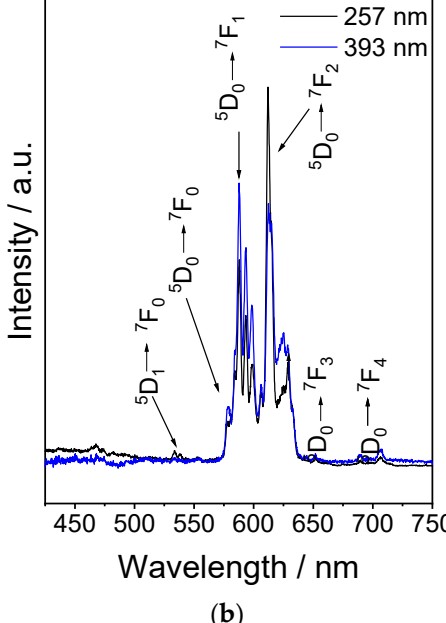

(**a**)             (**b**)

**Figure 3.** Luminescent properties of sol–gel derived $Y_2SiO_5$: $Eu^{3+}$ powders. (**a**) Excitation spectrum ($\lambda_{em}$ = 616 nm). (**b**) Emission spectra ($\lambda_{exc}$ = 257 and 393 nm).

Figure 3b shows the emission spectra obtained using with $\lambda_{exc}$ = 257 and 393 nm. For both excitation wavelengths, the same emissions are observed, and all bands associated with the intramolecular 4f $\longrightarrow$ 4f transitions of $Eu^{3+}$ are located in the red region of the electromagnetic spectrum, corresponding to the $^5D_0 \longrightarrow {}^7Fj$ (j = 0, 1, 2, 3, 4) transitions: $^5D_0 \longrightarrow {}^7F_0$ (578 nm), $^5D_0 \longrightarrow {}^7F_1$ (594 nm), $^5D_0 \longrightarrow {}^7F_2$ (616 nm), $^5D_0 \longrightarrow {}^7F_3$ (651 nm), and $^5D_0 \longrightarrow {}^7F_4$ (706 nm) [48]. Meanwhile, the small emission at 533 nm corresponds to the $^5D_1 \longrightarrow {}^7F_0$ transition. These results are consistent with the luminescent properties of $Eu^{3+}$ [49,50].

### 2.3.2. SPS-Sintered $Y_2SiO_5$: $Eu^{3+}$—Excitation Spectra

Considering that there may be a reduction of $Eu^{3+}$ to $Eu^{2+}$ during SPS, the excitation spectra were recorded for both $Eu^{3+}$ emission ($\lambda_{em}$ = 616 nm) and $Eu^{2+}$ emission ($\lambda_{em}$ = 433 nm). The results are shown in Figure 4a,b for 1.0 and 2.5 at.% Eu, for samples subjected to an SPS HT of 6 min of heat treatment in the SPS. The results are similar for both concentrations. In the case of $Eu^{3+}$, the same bands observed in the powders (Figure 3a) are present; that is, the charge transfer (CT) band from the $Y_2SiO_5$ host lattice to $Eu^{3+}$ appears between 220 and 320 nm, while the intense bands located around 390–391 nm are attributed to f-f transitions. Conversely, in the spectrum monitoring the blue emission at 433 nm, a broad excitation band from 240 to 400 nm is observed, which is attributed to the $4f^7(^8S_{7/2}) \longrightarrow 4f^65d^1$ transition of the $Eu^{2+}$ ion [51], confirming the reduction process from $Eu^{3+}$ to $Eu^{2+}$. The only observable difference regarding the change in concentration is that, for $Eu^{3+}$ ions, there is a lower intensity ratio of the f-f bands relative to the CT band at the higher Eu concentration (2.5 at.%). This suggests that increasing the Eu concentration enhances the probability of $Eu^{3+}$-$O^{2-}$ charge transfer and increases the likelihood of reduction to $Eu^{2+}$, which implies a lower number of intra-ionic *f-f* transitions due to the decreased content of $Eu^{3+}$.

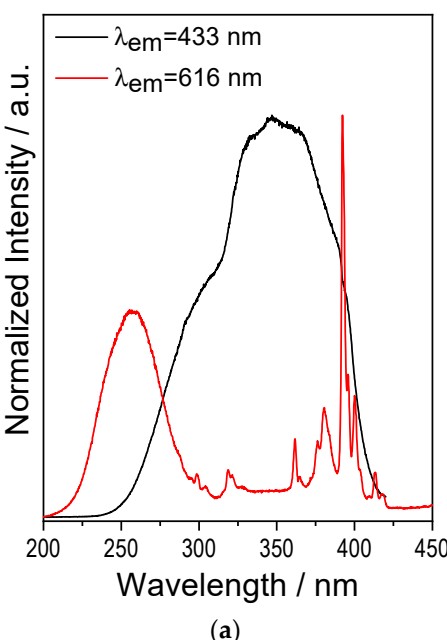

(a)

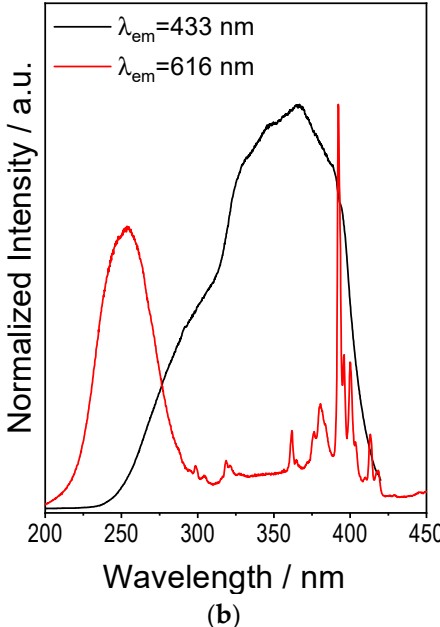

(b)

**Figure 4.** Excitation spectra of SPS-sintered $Y_2SiO_5$: $Eu^{3+}$ systems with $\lambda_{em}$ = 433 and 616 nm, HT = 6 min. (**a**) 1 at.% $Eu^{3+}$. (**b**) 2.5 at.% $Eu^{3+}$.

2.3.3. $Eu^{3+} \longrightarrow Eu^{2+}$ Reduction Process

The reduction of Eu is primarily attributed to the formation of carbon monoxide (CO), which is generated during the SPS when the graphite die reacts with residual oxygen at high temperatures, resulting in a low oxygen partial pressure [52,53]:

$$C + O_2 \longrightarrow CO \tag{1}$$

This CO acts as a reducing agent, first donating electrons to the system:

$$CO \longrightarrow CO^{2+} + 2e^- \tag{2}$$

These electrons then react with $Eu^{3+}$ as follows:

$$2Eu^{3+} + 2e^- \longrightarrow 2Eu^{2+} \tag{3}$$

Additionally, the rapid heating due to the Joule effect and the electric fields formed during SPS promote the formation of oxygen vacancies $V_{\ddot{O}}$ [54,55], which can form through the reaction of CO with lattice oxygen ($O_O^{2-}$) in the crystal structure:

$$CO + O_O^{2-} \longrightarrow CO_2 + \square_{\ddot{O}} + 2e^- \tag{4}$$

The overall reduction reaction can be expressed as:

$$2Eu^{3+} + CO + O_O^{2-} \longrightarrow 2Eu^{2+} + CO_2 + \square_{\ddot{O}} \tag{5}$$

In summary, because of the low oxygen partial pressure, CO is first formed, which donates a pair of electrons that reduce $Eu^{3+}$ to $Eu^{2+}$, followed by the formation of oxygen vacancies to maintain charge balance. Structurally, in Kröger–Vink notation, this can be expressed as:

$$2Eu_{Y'} + CO + O_O^x \longrightarrow 2Eu_{\dot{Y}} + CO_2 + \square_{\ddot{O}} \tag{6}$$

where $Eu_{Y'}' = Eu^{3+}$ at the $Y^{3+}$ site, $Eu_{\dot{Y}} = Eu^{2+}$ at the $Y^{3+}$ site, $O_O^x = O^{2-}$ at a regular lattice site, and, $\square_{\ddot{O}}$ = an oxygen vacancy.

2.3.4. SPS-Sintered $Y_2SiO_5$: $Eu^{3+}$—Emission Spectra

To analyse the luminescent properties as a function of Eu content and HT in the SPS process, samples were excited at three different wavelengths: 220, 257 and 365 nm. The selection of these wavelengths was based on the following criteria. The first excitation (220 nm) lies in a region where although a low luminescent response is obtained, only $Eu^{3+}$ emission is expected because there is no $Eu^{2+}$ response in this region, as shown in Figure 4.

The second excitation (257 nm) corresponds to the region where emissions from both cationic states overlap, and thus, both emissions should be observed. The third excitation (365 nm) preferentially corresponds to the emissive processes of $Eu^{2+}$. The results are shown in Figure 5.

For $\lambda_{exc}$ = 220 nm (Figure 5a,b for 1.0 and 2.5 at.% $Eu^{3+}$), only the emission bands associated with the intramolecular 4f $\longrightarrow$ 4f transitions of $Eu^{3+}$ located in the red region of the electromagnetic spectrum are observed, corresponding to the intramolecular $^5D_0 \longrightarrow ^7F\_j$ (j = 0, 1, 2, 3, 4) transitions: $^5D_0 \longrightarrow ^7F_0$ (578 nm), $^5D_0 \longrightarrow ^7F_1$ (594 nm), $^5D_0 \longrightarrow ^7F_2$ (616 nm), $^5D_0 \longrightarrow ^7F_3$ (651 nm), and $^5D_0 \longrightarrow ^7F_4$ (706 nm).

Additionally, it is possible to determine whether the HT modifies the local crystal field environment around the $Eu^{3+}$ ions in the host lattice by analysing the intensity ratio R of the $^5D_0 \longrightarrow ^7F_2$ to $^5D_0 \longrightarrow ^7F_1$ transitions, given that the $^5D_0 \longrightarrow ^7F_2$ and $^5D_0 \longrightarrow ^7F_1$

emissions correspond to electric dipole and magnetic dipole transitions, respectively. The former is a hypersensitive transition and thus highly sensitive to site symmetry, whereas the latter is unaffected by the symmetry of the crystal environment [56].

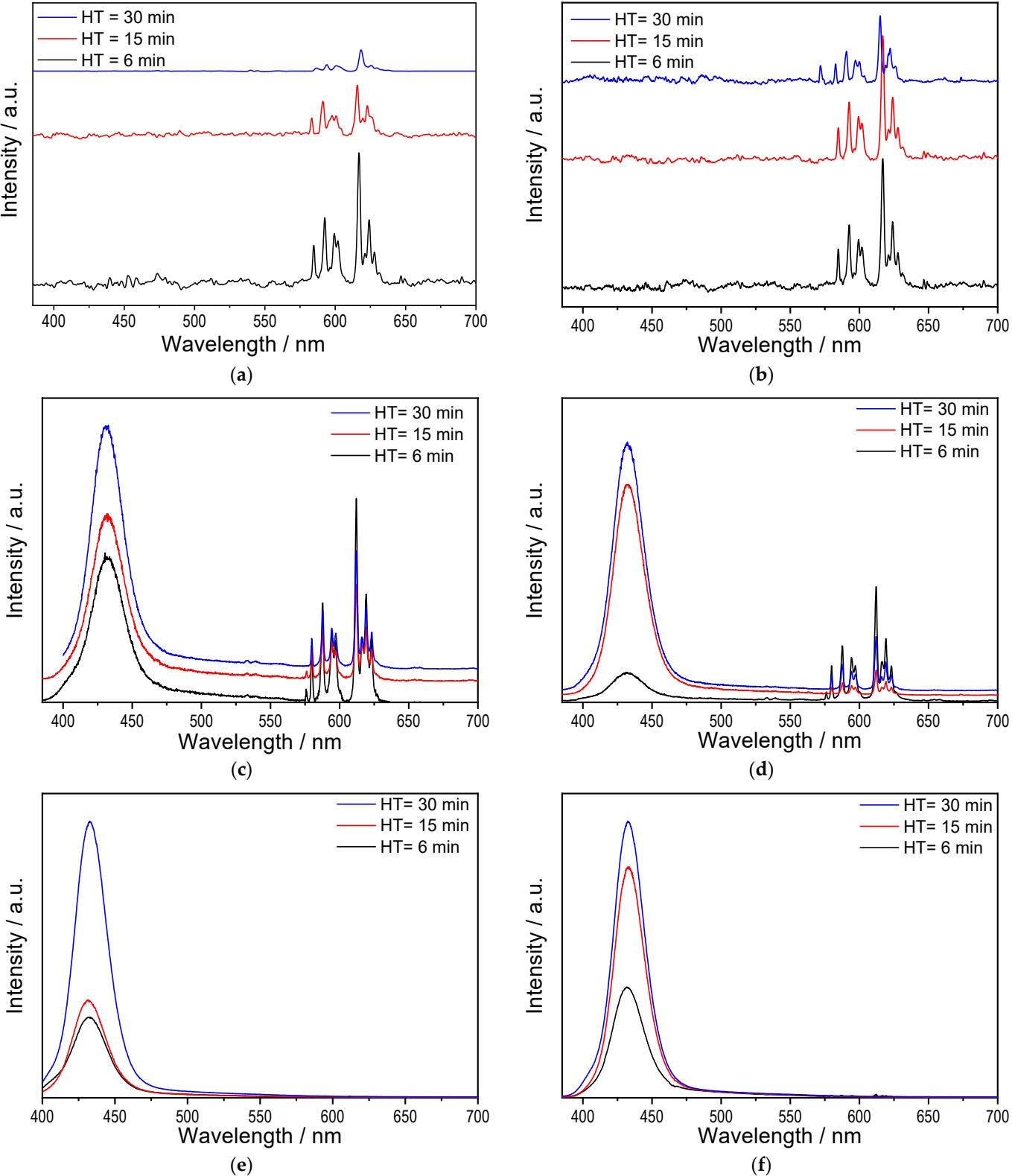

**Figure 5.** Emission spectra of SPS-sintered $Y_2SiO_5$: $Eu^{3+}$ samples: (**a**) 1.0 at.% Eu, $\lambda_{exc}$ = 220 nm; (**b**) 2.5 at.% Eu, $\lambda_{exc}$ = 220 nm; (**c**) 1.0 at.% Eu, $\lambda_{exc}$ = 257 nm; (**d**) 2.5 at.% Eu, $\lambda_{exc}$ = 257 nm; (**e**) 1.0 at.% Eu, $\lambda_{exc}$ = 365 nm; (**f**) 2.5 at.% Eu, $\lambda_{exc}$ =365 nm.

The results demonstrate that increasing the HT in the SPS chamber (from 6 to 15 to 30 min) leads to a decrease in the R value, from 1.98 to 1.08 to 1.06 for 1.0 at.% Eu, and from 2.06 to 1.32 to 1.15 for 2.5 at.% Eu, recall that $X_2$-$Y_2SiO_5$ presents two sites with different coordination numbers (CN), namely 6 and 7, with the latter being more asymmetric. At 6 min, the R value indicates that $Eu^{3+}$ ions preferentially occupy the more asymmetric sites (CN = 7). By 30 min, the remaining $Eu^{3+}$ ions are preferentially located at the more symmetric sites (CN = 6).

This implies that the CN = 7 sites are reduced first, likely because the bonds are slightly longer (average values of 2.367 and 2.329 Å for CN = 7 and 6, respectively; see Table 1), making them weaker and easier to break. Additionally, the CN = 7 sites exhibit lower packing density (owing to their distorted pentagonal bipyramidal geometry) compared with the CN = 6 sites (which have an almost regular octahedral geometry), allowing CO molecules to approach the Eu ions more easily, facilitating their reduction.

Regarding the samples excited at $\lambda_{exc}$ = 257 nm thus exciting both $Eu^{3+}$ and $Eu^{2+}$, the emission from both cations is indeed observed: those previously described for $Eu^{3+}$ (between 550 and 650 nm) and a band centred at 423 nm corresponding to the $4f^65d^1 \longrightarrow 4f^7$ electronic transitions resulting from parity-allowed transitions of $Eu^{2+}$ [57,58]. The results demonstrate that Eu reduction has indeed been achieved.

Moreover, as the HT increases, the emission band of $Eu^{2+}$ also increases, while the emission bands of $Eu^{3+}$ decrease, demonstrating that a longer HT leads to greater Eu reduction for both 1.0 and 2.5 at.% Eu.

Finally, when $\lambda_{exc}$ = 365 nm (Figure 5e,f for 1.0 and 2.5 at.% Eu), only the emission from $Eu^{2+}$ is observed, which increases as HT increases. This emission band can be attributed to the $^5D_3 \longrightarrow {}^7F_1$ transition of $Eu^{3+}$ [59], and thus, the emission centred at 423 nm in Figure 5 corresponds to the reduced $Eu^{2+}$. Comparing Figure 5c,d, as well as Figure 5e,f, the reduction kinetics differ for the samples with different Eu contents. The Eu reduction rate is greater in the sample with 2.5 at.% because the Eu content is higher, kinetically increasing the probability that the reduction reaction will occur.

To analyse the nature of the emission originating from $Eu^{2+}$, only the blue region of the spectrum was examined for the sample with 2.5 at.% Eu as a function of HT (Figure 6a). The emission intensity increases with the HT, demonstrating greater reduction of the cation.

Nevertheless, when HT = 6 min, the observed emission consists of multiple bands. To analyse them, a Gaussian deconvolution of the corresponding spectrum was carried out (Figure 6b). For the deconvolution of PL spectra into individual components and to avoid misinterpretation [60], the spectra were plotted as a function of energy using Planck's conversion, transposing the y-axis by a factor of $\lambda^2$/hc (the Jacobian transformation) [61].

In this manner, the band can be decomposed into seven components, ranging from 2.75 to 3.17 eV. For $4f^65d^1 \longrightarrow 4f^7$ electronic transitions, the band position is characteristic of specific materials, because the position of the 5d levels strongly depends on the crystal environment surrounding $Eu^{2+}$. Variations in the energy levels between the excited and ground states can differ by tens of thousands of $cm^{-1}$ for the $Eu^{2+}$ ion [62], meaning that the emission band position strongly depends on the surroundings of the luminescent centre within the crystal structure.

For example, in $YSiO_2N$ doped with $Eu^{2+}$, the emission ranges from 550 to 1100 nm [63], whereas in (La-Al)$_2O_3$: $Eu^{2+}$, the band is centred at 440 nm [64], and in $Sr_2Si_5N_8$: $Eu^{2+}$, at 620 nm [65]. Thus, emission may be observed across a wide range, from 330 to 1100 nm.

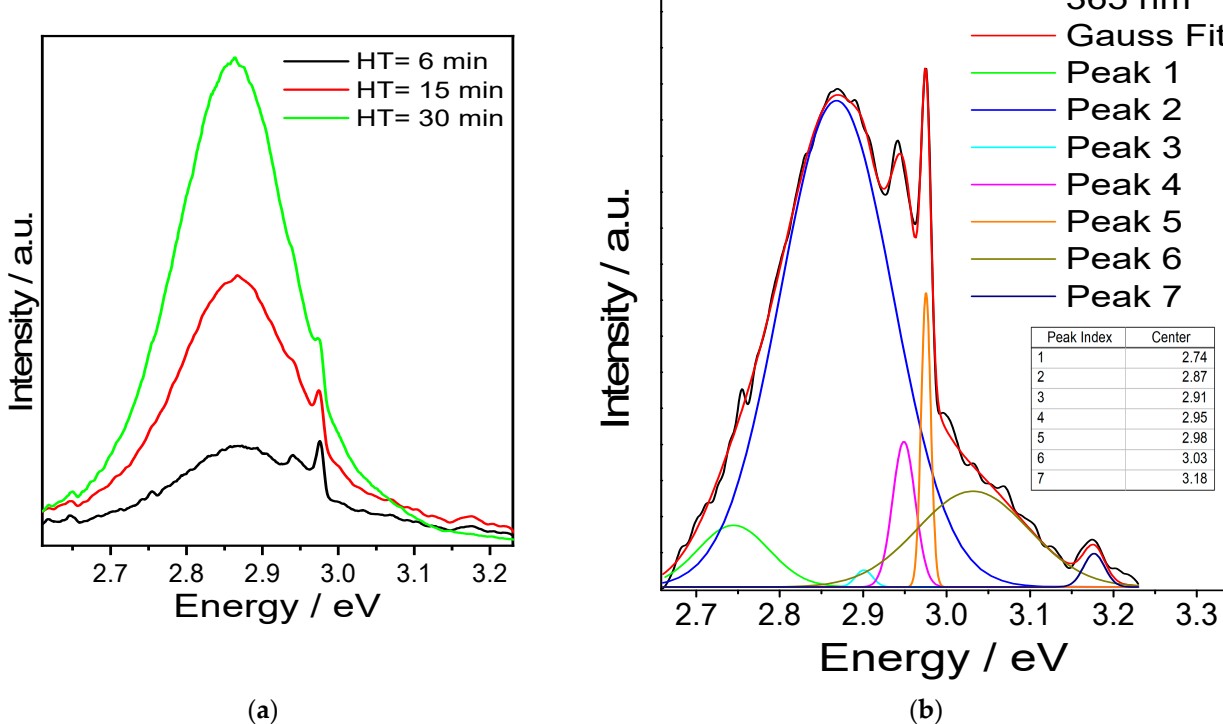

**Figure 6.** (**a**) Emission spectrum for SPS-sintered $Y_2SiO_5$ samples doped with 2.5 at.% Eu, blue region, $\lambda_{exc}$ = 365 nm. (**b**) Gaussian deconvolution, HT = 6 min.

To predict the emission position of $Eu^{2+}$ based on the crystal structure, Van Uitert et al. [66] analysed various crystal structures and determined the following empirical relationship (Equation (7)):

$$E = Q\left[1 - \left(\frac{V}{4}\right)^{1/V} \times 10^{-\left(\frac{n \cdot r \cdot ea}{80}\right)}\right] \tag{1}$$

where E = the energy position of the $Eu^{2+}$ emission peak, Q = is the energy position of the lower edge of the d-band for the free $Eu^{2+}$ ion (Q = 34,000 $cm^{-1}$), that is, the minimum energy of the d-electron level, V = is the valence of the $Eu^{2+}$ ion (V = 2), n = is the number of anions in the immediate coordination shell around the $Eu^{2+}$ ion, (i.e., the CN), ea = is the electron affinity of the atoms forming the anions bonded to $Eu^{2+}$ (in eV), and r = is the radius of the host cation replaced by the $Eu^{2+}$ ion (in Å).

In the present study, it is assumed that r corresponds to the average bond length between yttrium ions and oxygen atoms because the $Eu^{2+}$ cation is considered to substitute yttrium at these sites. For this purpose, the bond distances calculated by Mirzai et al. [67] for the $X_2$-$Y_2SiO_5$ structure are used.

For the two sites, Y1 and Y2, (with CN = 7 and 6, respectively), there are two different types of bonds: in Y1, there are 5 bonds directly to $SiO_4^{4-}$ oxy-anions and 2 to $O^{2-}$, while in Y2, there are 4 of the first type and 2 of the second type. Therefore, for the electron affinity, two possible paths for energy transfer exists: one from the Y–$SiO_4$ bond, with ea = 4.08 eV [68], and another for the Y–O bond with ea = 1.89 eV [69,70].

The results are shown in Table 1. As observed, the predicted band positions exhibit a high correlation with experimental values. The first observation is that the blue emission apparently corresponds to energy transfer through the $SiO_4^{4-}$ oxy-anions for both

crystalline sites, demonstrating the importance of the local environment surrounding $Eu^{2+}$, since emissions from the Eu–O bond are not observed.

**Table 1.** $Eu^{2+}$ emission bands predicted using the Van Uitert Equation.

| Site | Coordination Number | Bond | Bond Length/Å [67] | λ/nm | Energy/eV |
|------|---------------------|------|--------------------|------|-----------|
| Y1 | 7 | Y-O | 0.2239 | 598.66 | 2.07 |
| | | Y-O | 0.2367 | 587.32 | 2.11 |
| | | $Y-SiO_4$ | 0.2379 | 400.34 | 3.10 |
| | | $Y-SiO_4$ | 0.2589 | 388.71 | 3.19 |
| | | $Y-SiO_4$ | 0.3290 | 359.69 | 3.45 |
| | | $Y-SiO_4$ | 0.2329 | 403.37 | 3.07 |
| | | $Y-SiO_4$ | 0.2337 | 402.88 | 3.08 |
| Y2 | 6 | Y-O | 0.2209 | 632.83 | 1.96 |
| | | Y-O | 0.2325 | 622.42 | 1.99 |
| | | $Y-SiO_4$ | 0.2392 | 422.47 | 2.94 |
| | | $Y-SiO_4$ | 0.2296 | 428.93 | 2.89 |
| | | $Y-SiO_4$ | 0.2358 | 424.71 | 2.92 |
| | | $Y-SiO_4$ | 0.2394 | 422.34 | 2.94 |

Regarding the position of the luminescent centres, the results indicate that the higher-energy emissions correspond to the CN = 7 site, and the lower-energy emissions correspond to the CN = 6 site. Notably at an initial HT = 6 min, the most intense emission originates from the CN = 7 sites, demonstrating that these are the first to be reduced, confirming the results observed with $\lambda_{exc}$ = 220 nm. Furthermore, as the HT increases, the emission from the CN = 6 sites increases, so that ultimately these sites become predominant.

2.3.5. SPS-Sintered $Y_2SiO_5$: $Eu^{3+}$—Lifetime Studies

Regarding the lifetime (τ) analysis, all samples were monitored in two experiments: with $\lambda_{exc}$ = 257 nm and $\lambda_{em}$ = 616 nm for $Eu^{3+}$ emission; and with $\lambda_{exc}$ = 365 nm and $\lambda_{em}$ = 433 nm for $Eu^{2+}$ emission. For $Eu^{3+}$, the results were adjusted using the first-order kinetics or monomolecular reaction mechanism (Equation (8)):

$$I(t) = Ae^{-t/\tau} \tag{8}$$

where I(t) is the intensity at a given time t, A is the intensity at t = 0, and τ is the lifetime. The fit is shown in Figure 7a,b for 1.0 and 2.5 at.%, and the lifetime values are presented in Table 2. For $Eu^{2+}$, the best fitting of the experimental data was obtained using the double exponential equation (Equation (9)) [67]:

$$I(t) = I_0 + A_1 e^{-t/\tau_1} + A_2 e^{-t/\tau_2} \tag{9}$$

where I(t) is the luminescence intensity at time t, $A_1$ and $A_2$ are constants, and $\tau_1$ and $\tau_2$ are rapid and slow times for the exponential components, respectively. From Equation (4), the average lifetime $\tau_{av}$ can be calculated (Equation (10)):

$$\tau_{av} = \frac{A_1\tau_1^2 + A_2\tau_2^2}{A_1\tau_1 + A_2\tau_2} \tag{10}$$

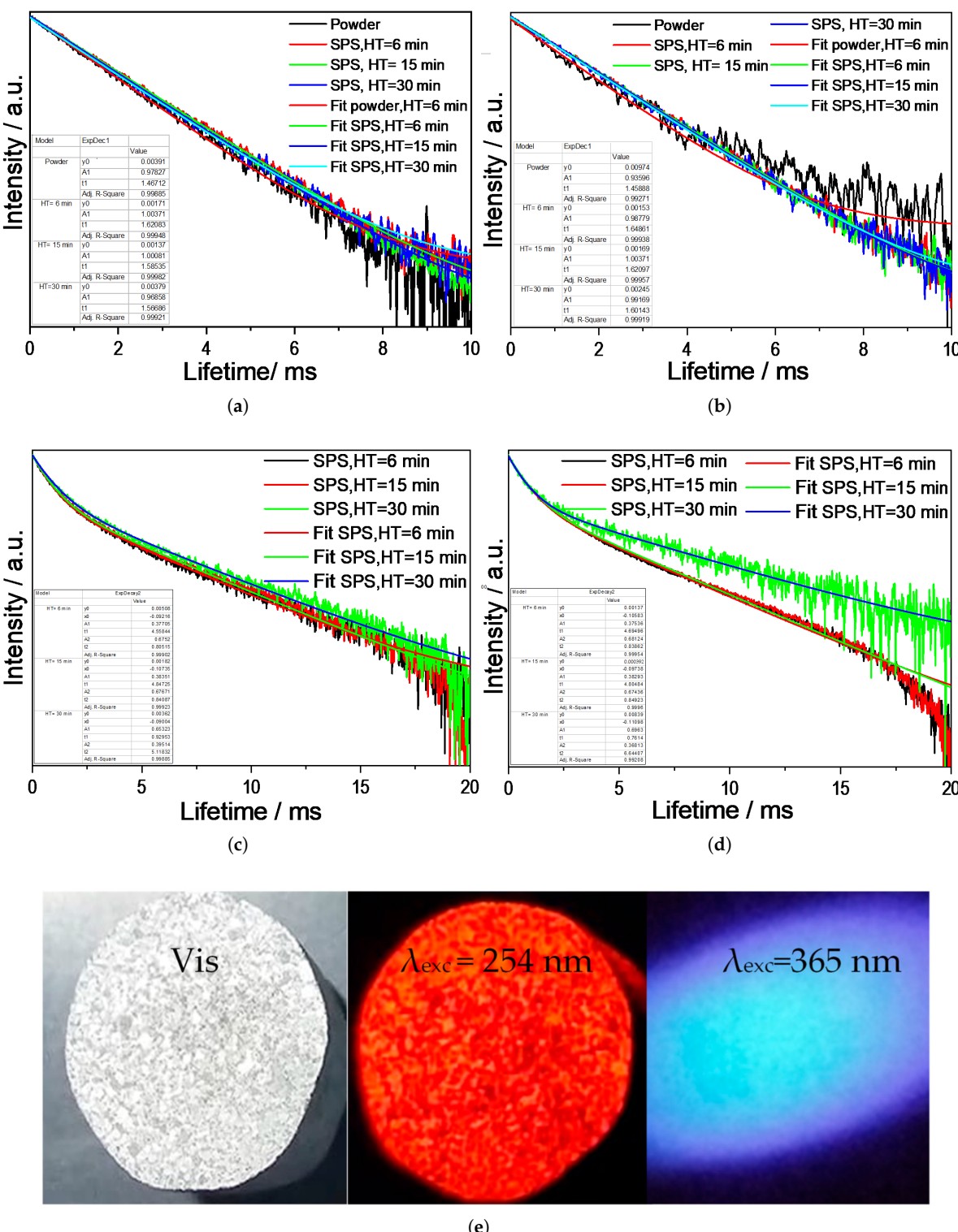

**Figure 7.** Average lifetime for sintered $Y_2SiO_5$ as a function of holding time (HT). (**a**) 1.0 at.% Eu, $\lambda_{exc}$ = 257 nm, $\lambda_{em}$ = 616 nm. (**b**) 2.5 at.% Eu, $\lambda_{exc}$ = 257 nm, $\lambda_{em}$ = 616 nm. (**c**) 1.0 at.% Eu, $\lambda_{exc}$ = 365 nm, $\lambda_{em}$ = 433 nm. (**d**) 2.5 at.% Eu, $\lambda_{exc}$ = 254 nm and $\lambda_{em}$ = 365 nm, HT = 6 min. (**e**) 2.5 at.% Eu, under natural light, $\lambda_{exc}$ = 254 nm and $\lambda_{exc}$ = 365 nm.

The fitting is shown in Figure 7c,d for 1.0 and 2.5 at.% Eu, and the obtained values are presented in Table 3. The decay of the $Eu^{3+}$ emission follows a first-order model because its $^5D$ level is relatively isolated, considering that the f-f transitions limit non-radiative de-excitation pathways. Additionally, there is a large energy gap with the ground state, which

minimises multiphonon quenching mechanisms; hence, the de-excitation mechanisms are essentially radiative.

**Table 2.** Lifetime for $Eu^{3+}$.

| Eu Content/at.% | HT/min | $\tau$/ms | $R^2$ |
|---|---|---|---|
| 1.0 | Powder | 1.4671 | 0.9988 |
| | 6 | 1.6208 | 0.9994 |
| | 15 | 1.5853 | 0.9998 |
| | 30 | 1.5668 | 0.9992 |
| 2.5 | Powder | 1.4588 | 0.9927 |
| | 6 | 1.6486 | 0.9993 |
| | 15 | 1.6209 | 0.9995 |
| | 30 | 1.6014 | 0.9991 |

**Table 3.** Lifetime for $Eu^{2+}$.

| Eu Content/at.% | HT/min | $\tau_1$/ms | $\tau_2$/ms | $\tau_{avg}$/ms | $R^2$ |
|---|---|---|---|---|---|
| 1.0 | 6 | 4.5584 | 0.8051 | 3.6565 | 0.9990 |
| | 15 | 4.8472 | 0.8408 | 3.9083 | 0.9992 |
| | 30 | 0.9295 | 5.1183 | 4.1511 | 0.9988 |
| 2.5 | 6 | 4.6949 | 0.8386 | 3.7508 | 0.9996 |
| | 15 | 4.8048 | 0.8492 | 3.8658 | 0.9995 |
| | 30 | 0.7614 | 6.644 | 5.6114 | 0.9920 |

The situation for $Eu^{2+}$ is more complex because its $4f^65d \longrightarrow 4f^7$ transition is allowed and very fast, but the 5d state is coupled to the crystal lattice. This creates two effects: (1) the parabolic potential well allows vibrational relaxation towards the minimum, and (2) the thermal barrier produces repopulation of the excited state. This thermal equilibrium between excited states requires the modelling of two coupled decay pathways. Moreover, the 5d state is more spatially extended than the 4f orbitals of $Eu^{3+}$, making ion–ion interactions more likely, which explains the observed biexponential decay behaviour.

From the calculated data the decay time of $Eu^{3+}$ decreases as the HT increases, which is consistent with the fact that a higher HT leads to a lower concentration of $Eu^{3+}$ cations in the system, and that $Eu^{2+}$ cations may act as quenching centres. Additionally, increasing sintering promotes the formation of oxygen vacancies, which serve as charge trapping centres. Furthermore, the increase in decay time from the unsintered powder to the sintered samples is primarily due to the structural change from X1 to X2, as well as the reduction in surface defects and increased crystallinity, which diminishes non-radiative pathways. A slight increase in $\tau$ is also expected from 1.0 to 2.5 at.% Eu owing to the higher cation content.

Regarding $Eu^{2+}$, the average lifetime ($\tau_{avg}$) increases with HT, suggesting an increase in luminescent efficiency, probably because $Eu^{2+}$ ions migrate to more favourable crystallographic sites in the $Y_2SiO_5$ lattice (i.e., sites with lower local symmetry that reduce vibrational coupling) and thereby decrease the probability of non-radiative de-excitation.

Notably, the fast component $\tau_1$ significantly decreases for both Eu concentrations, whereas the slow component $\tau_2$ increases, suggesting a change in mechanism, the fast component ($\tau_1$) may involve stronger quenching by defects, whereas the slow component

($\tau_2$) may reflect the formation of protective environments. This also suggests that $Eu^{2+}$ migrates to environments with lower local symmetry, which correspond to the Y2 sites in this case. This is consistent with the luminescence results, which demonstrated that after 30 min, the predominant emission originates from these sites.

2.3.6. SPS-Sintered $Y_2SiO_5$: $Eu^{3+}$—CIE Coordinates

Table 4 and Figure 8 show the CIE coordinates for the sintered systems; different colours are obtained depending on the excitation wavelength and the HT. Excitation at $\lambda exc = 220$ nm produces colours in the red region (originating from $Eu^{3+}$), excitation at $\lambda exc = 365$ nm produces colours in the blue region (originating from $Eu^{2+}$), and excitation at $\lambda exc = 257$ nm results in a mixture of both colours. Therefore, it is possible to obtain colours along the lines between points D and G for 1.0 at.%, and between points O and R for 2.5 at.%, depending on the HT used during SPS.

**Table 4.** CIE chromaticity coordinates for the $Y_2SiO_5$: Eu system before and after SPS as a function of HT.

| Point | Eu Content/at.% | HT/min | $\lambda_{exc}$/nm | x | y | CCT/K | R | G | B | DW/nm | CP | HEX |
|---|---|---|---|---|---|---|---|---|---|---|---|---|
| A | | 6 | | 0.5846 | 0.3742 | 1414 | 255 | 91 | 0 | 598.4 | 87.8 | #FF5B00 |
| B | | 15 | 220 | 0.5700 | 0.3492 | 1366 | 255 | 82 | 56 | 604.3 | 75.8 | #FF5238 |
| C | | 30 | | 0.5497 | 0.3357 | 1403 | 254 | 82 | 75 | 609.1 | 65.7 | #FE524B |
| D | | Powder | | 0.6007 | 0.3769 | 1348 | 254 | 86 | 0 | 598.4 | 93.4 | #FE5600 |
| E | | 6 | 257 | 0.2741 | 0.1131 | - | 208 | 0 | 255 | N/A | N/A | #D000FF |
| F | 1.0 | 15 | | 0.2254 | 0.0918 | - | 157 | 0 | 255 | N/A | N/A | #9D00FF |
| G | | 30 | | 0.2148 | 0.0785 | - | 147 | 0 | 255 | N/A | N/A | #9300FF |
| H | | Powder | | 0.3303 | 0.2316 | - | 255 | 140 | 240 | N/A | N/A | #FF8CF0 |
| I | | 6 | 365 | 0.1646 | 0.0430 | - | 83 | 0 | 255 | 454.0 | 93.1 | #5300FF |
| J | | 15 | | 0.1645 | 0.0327 | - | 88 | 0 | 255 | 450.4 | 95.3 | #5800FF |
| K | | 30 | | 0.1649 | 0.0284 | - | 90 | 0 | 255 | 448.4 | 96.2 | #5A00FF |
| L | | 6 | | 0.6338 | 0.3472 | 1080 | 255 | 47 | 0 | 605.9 | 94.4 | #FF2F00 |
| M | | 15 | 220 | 0.5886 | 0.3574 | 1314 | 255 | 79 | 35 | 602.4 | 83.9 | #FF4F23 |
| N | | 30 | | 0.5212 | 0.3211 | 1488 | 255 | 86 | 96 | 619.1 | 52.7 | #FF5660 |
| O | | Powder | | 0.6035 | 0.3646 | 1280 | 255 | 77 | 0 | 601.1 | 90.5 | #FF4D00 |
| P | | 6 | 257 | 0.3989 | 0.2277 | - | 255 | 78 | 184 | N/A | N/A | #FF4D00 |
| Q | 2.5 | 15 | | 0.1786 | 0.0450 | - | 107 | 0 | 255 | 444.3 | 90.1 | #6B00FF |
| R | | 30 | | 0.1915 | 0.0529 | - | 123 | 0 | 255 | 431.2 | 86.0 | #7B00FF |
| S | | Powder | | 0.3608 | 0.2399 | - | 255 | 121 | 210 | N/A | N/A | #FF79D2 |
| T | | 6 | 365 | 0.1708 | 0.0535 | - | 90 | 0 | 255 | 454.0 | 89.8 | #5A00FF |
| U | | 15 | | 0.1648 | 0.0304 | - | 89 | 0 | 255 | 449.2 | 95.8 | #5900FF |
| V | | 30 | | 0.1649 | 0.0284 | - | 90 | 0 | 255 | 448.4 | 96.2 | #5A00FF |

Moreover, the longer the HT, the more the colours shift toward blue. At the same time, for $\lambda exc = 220$ nm, it is observed that prolonged sintering enhances the reduction of $Eu^{3+} \longrightarrow Eu^{2+}$, increasing the $Eu^{2+}$ concentration and broadening its emission band. This reduces colour purity and shifts the emission toward longer wavelengths.

Comparing Figure 8b,e (as well as the corresponding CIE coordinates), the sample with lower Eu content (1.0 at.%) shows a relatively rapid shift towards the blue region of the spectrum, meaning that there is greater efficiency in the reduction mechanisms. Conversely, the sample with the higher Eu content (2.5 at.%) shows lower efficiency because of the mixture between $Eu^{3+}$ and $Eu^{2+}$, allowing light to be obtained in the white region of the spectrum after HT = 15 min.

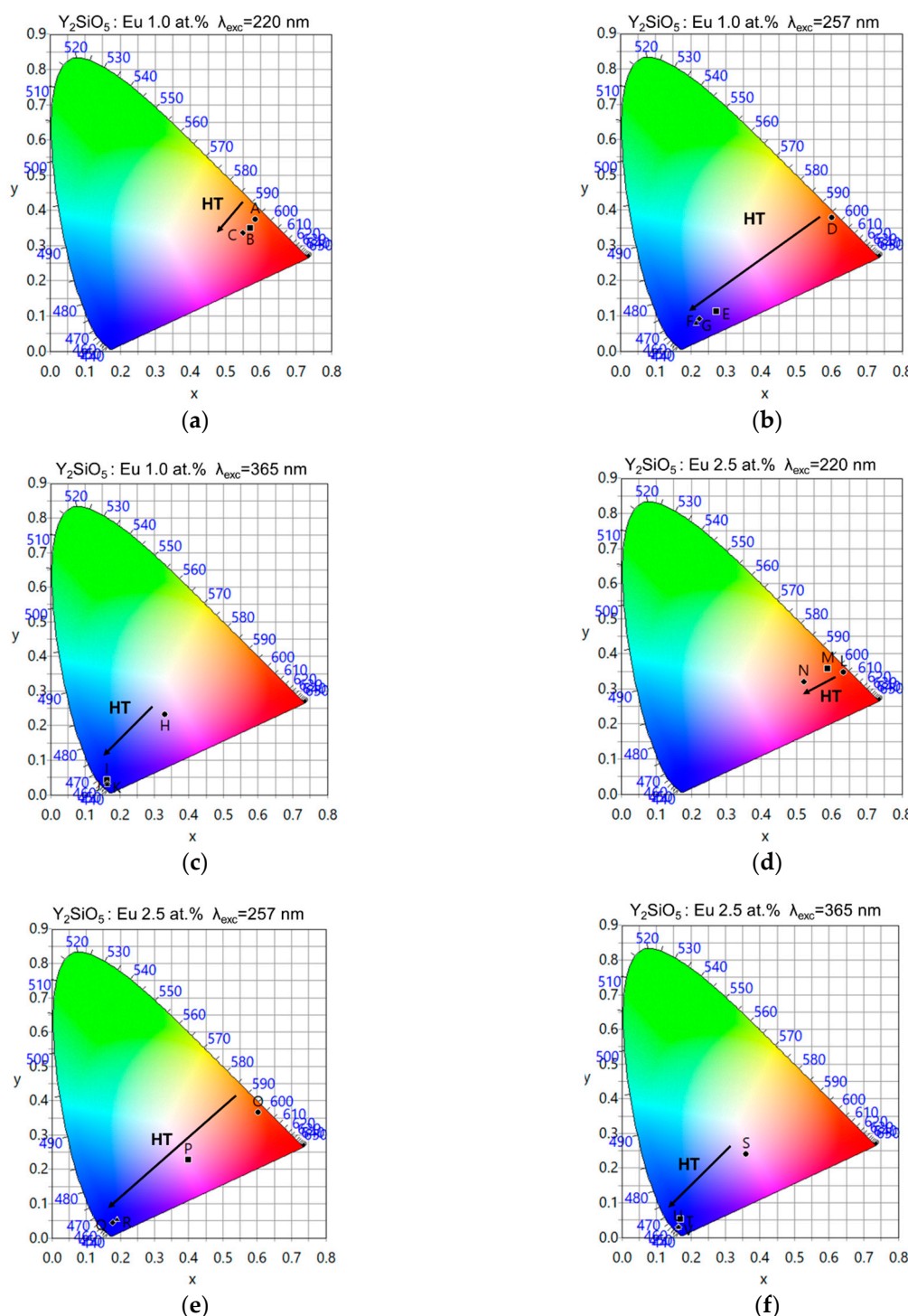

**Figure 8.** CIE coordinates for sintered $Y_2SiO_5$, as function of HT. (**a**) 1.0 at.% Eu, $\lambda_{exc}$ = 220 nm. (**b**) 1.0 at.% Eu, $\lambda_{exc}$ = 257 nm. (**c**) 1.0 at.% Eu, $\lambda_{exc}$ = 365 nm. (**d**) 2.5 at.% Eu, $\lambda_{exc}$ = 220 nm, (**e**) 2.5 at.% Eu, $\lambda_{exc}$ = 257 nm. (**f**) 2.5 at.% Eu, $\lambda_{exc}$ = 365 nm.

### 2.3.7. Oxidation Process of SPS-Sintered $Y_2SiO_5$: $Eu^3$—Luminiscent Properties

The $Y_2SiO_5$: Eu samples sintered for HT = 30 min with 1.0 and 2.5 at.% Eu were placed in a furnace for 1 h at 1000 °C in an uncontrolled (ambient) atmosphere, aiming to re-oxidise $Eu^{2+}$ back to $Eu^{3+}$. The luminescence results for the oxidised samples are shown in Figure 9.

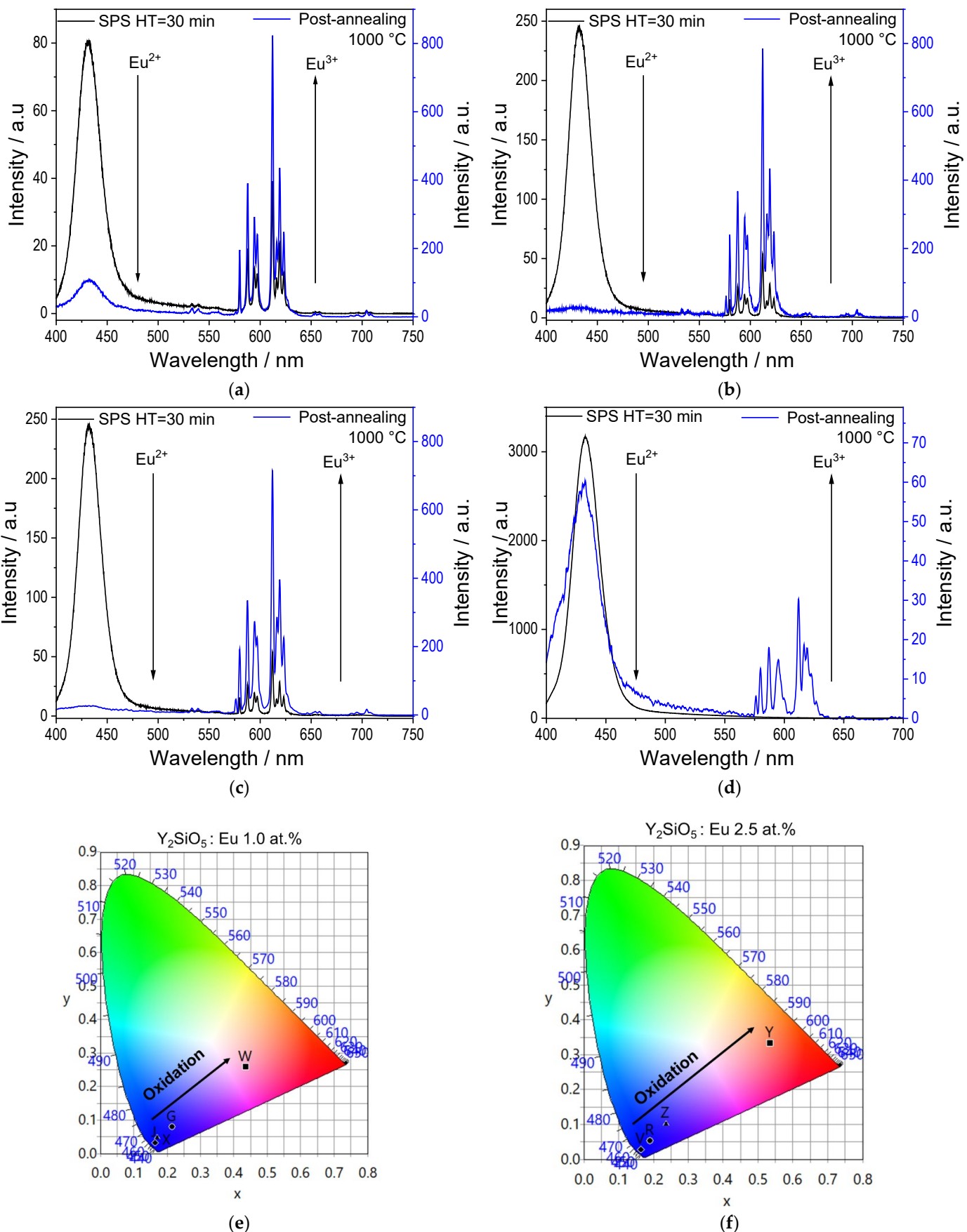

**Figure 9.** Luminescent properties of sintered $Y_2SiO_5$: Eu samples after oxidation process at 1000 °C. (**a**) $\lambda_{exc}$ = 257 nm, 1.0 at.%. Eu. (**b**) $\lambda_{exc}$ = 257 nm, 2.5 at.% Eu. (**c**) $\lambda_{exc}$ = 365 nm, 1.0 at.%. Eu. (**d**) $\lambda_{exc}$ = 365 nm, 2.5 at.% Eu. (**e**) CIE coordinates for 1.0 at.% Eu. (**f**) CIE coordinates for 2.5 at.% Eu.

For both Eu concentrations, the emission corresponding to $Eu^{2+}$ at 433 nm significantly decreases after oxidation at 1000 °C (the emission spectra of the samples after sintering are included for ease of comparison). This demonstrates that the $Eu^{2+} \longrightarrow Eu^{3+}$ process has occurred. This result is evident in points W and Y of the CIE diagrams (Figure 9e,f, and Table 5); at $\lambda_{exc}$ = 257 nm, the colours shift from blue back towards red, demonstrating that the process is reversible. Finally, at $\lambda_{exc}$ = 365 nm, a decrease in the $Eu^{2+}$ emission band is also observed; however, from the emission spectra, a certain amount of $Eu^{2+}$ remains.

**Table 5.** CIE chromaticity coordinates for the $Y_2SiO_5$: Eu system before and after the oxidation at 1000 °C process.

| Point | Eu Content/at.% | HT/min | $\lambda_{exc}$/nm | x | y | CCT/K | R | G | B | DW/nm | CP | HEX |
|---|---|---|---|---|---|---|---|---|---|---|---|---|
| G | | 30 | 257 | 0.2148 | 0.0785 | | 147 | 0 | 255 | N/A | N/A | #9300FF |
| W | 1.0 | Oxidised | | 0.4363 | 0.2590 | | 255 | 84 | 156 | N/A | N/A | #FF549C |
| J | | 30 | 365 | 0.1645 | 0.0327 | | 88 | 0 | 255 | 450.4 | 95.3 | #5800FF |
| X | | Oxidised | | 0.1711 | 0.0487 | | 93 | 0 | 255 | 452.0 | 90.7 | #5D00FF |
| R | | 30 | 257 | 0.1915 | 0.0529 | | 123 | 0 | 255 | 431.2 | 86.0 | #7B00FF |
| Y | 2.5 | Oxidised | | 0.5359 | 0.3325 | | 254 | 87 | 84 | 610.7 | 60.6 | #FE5754 |
| V | | 30 | 365 | 0.1649 | 0.0284 | 1464 | 90 | 0 | 255 | 448.4 | 96.2 | #5A00FF |
| Z | | Oxidised | | 0.2363 | 0.1018 | | 168 | 0 | 255 | N/A | N/A | #A800FF |

## 3. Materials and Methods

### 3.1. $Y_2SiO_5$ Powders Preparation

$Eu^{3+}$-doped $Y_2SiO_5$ powders were synthesised via sol–gel method. The process involved two precursor solutions: (I) Rare-earth sol and (II) Silicon sol. Solution (I) was prepared by dissolving yttrium chloride ($YCl_3 \cdot 6H_2O$, Sigma-Aldrich, St. Louis, MO, USA, 99.99%) in a 2:1 ethanol-ethylene glycol mixture ($C_2H_6O$, Fermont, Monterrey, Mexico, 98%; $C_2H_6O_2$, Sigma-Aldrich, 98%) to achieve a 0.23 M $Y^{3+}$ concentration. Acetic acid ($CH_3COOH$, Fermont, Monterrey, Mexico, 98%) was added as a catalyst at a concentration of 1.74 M. Europium (III) chloride ($EuCl_3 \cdot 6H_2O$, Sigma-Aldrich, St. Louis, MO, USA, 98%) was then introduced to obtain $Eu^{3+}$ doping concentrations of 1.0 and 2.5 at.%. The choice of the Eu content analysed was made considering that the maximum emission efficiency concentration of $Eu^{3+}$ is usually between 1 and 2.5 at.% for different ceramic systems [71,72] and for $Y_2SiO_5$ [73]. The solution was stirred for 4 h. Solution (II) was prepared with a 0.12 M Si concentration using tetraethyl orthosilicate (TEOS, $SiC_8H_{20}O_4$, Sigma-Aldrich, St. Louis, MO, USA, 99%) in an 8:1 ethanol-water mixture. Acetic acid (0.26 M) was added as a catalyst, and the solution was stirred for 4 h. After stirring, both sols were mixed and agitated for 24 h. The resulting sol was dried at 100 °C for 48 h to form a xerogel. Subsequently, thermal treatments were performed at 300 °C and 500 °C for 2 h each to remove organic residues. Finally, the powder was calcined at 1000 °C for 4 h to achieve complete crystallisation into $Y_2SiO_5$: $Eu^{3+}$.

### 3.2. $Y_2SiO_5$: $Eu^{3+}$ Sintering and Later Annealing

$Y_2SiO_5$: $Eu^{3+}$ sol–gel powders were sintered using a Dr Sinter Sumitomo 1050 apparatus (Sumitomo Coal Ming Co., Tokyo, Japan), 2 g was weighed and placed in a 5 mm diameter graphite die with a graphite foil lining. The sintering was carried out at temperature of 1300 °C with uniaxial pressure of 4.5 kN (28.645 MPa). The pressure was applied at the beginning of the heating cycle at a rate of 100 °C/min. For both two Eu contents (1.0 and 2.5 at.%), samples were prepared with three different holding times: 6, 15 and 30 min. After holding this time, the ram pressure was released, and the solid mixture was allowed

to cool inside the chamber. In this way, coupons of 5 mm diameter and 10 mm thick were obtained. Once the samples were characterised, they were subjected to a heat treatment at 1000 °C for 1 h in air, in order to oxidise the Eu again. On the other hand, to ensure the correct compaction process, shrinkage is monitored as a function of both temperature and time. Shrinkage can be defined as: $dL/L_0$, where $L_0$ is the initial thickness of the specimen, and is therefore considered to be the instantaneous displacement rate.

### 3.3. Characterisation

Crystal structure was determined by a D2 Phase-Bruker diffractometer (Karlsruhe, Germany) using a copper anticathode at 40 kV and 20 mA. Luminescent properties were analysed with an Acton Pro 3500i monochromator (Acton Research Corporation, Acton, MA, USA) and a R955 Hamamatsu photomultiplier tube (Shizuoka, Japan) for visible emission with a fluorescence spectrophotometer Hitachi F-7000 (Ibaraki, Japan), equipped with a 150 W xenon lamp.

## 4. Conclusions

We demonstrated that sol–gel-derived $Y_2SiO_5$ powders doped with $Eu^{3+}$, when processed by SPS, exhibit changes in the luminescent properties owing to the reduction of $Eu^{3+} \longrightarrow Eu^{2+}$. This phenomenon depends on the HT during SPS, showing that the longer the sintering time, the greater the amount of reduced Eu.

Cation reduction occurs because of the synergistic action between CO inside the chamber and the creation of oxygen vacancies, facilitating the formation of $Eu^{2+}$. Therefore, the luminescent properties of the systems can be tuned depending on the initial Eu content and the SPS HT, as well as the excitation wavelength.

Ultimately, excitation at 220 nm always produces the typical red colours of $Eu^{3+}$, whereas excitation at 365 nm produces the blue colours of $Eu^{2+}$, and a mixture of both colours is observed at 257 nm, specifically blues, pinks, purples, whites, oranges, and reds. Finally, we demonstrated that the reduction process is reversible ($Eu^{2+} \longrightarrow Eu^{3+}$), so the luminescent properties can be tuned by controlling the chemical composition, the sintering time in the SPS and the oxidation process.

**Author Contributions:** Conceptualization, F.J.-L. and A.d.J.M.-R.; methodology, M.A.N.-V.; validation, M.A.N.-V., R.C.-M. and M.J.S.-M.; formal analysis, A.d.J.M.-R. and M.G.-H.; investigation, F.J.-L.; writing—original draft preparation, M.A.N.-V. and M.J.S.-M.; writing—review and editing, A.d.J.M.-R. and M.G.-H.; visualisation, R.C.-M. All authors have read and agreed to the published version of the manuscript.

**Funding:** This research received no external funding.

**Institutional Review Board Statement:** Not applicable.

**Informed Consent Statement:** Not applicable.

**Data Availability Statement:** The original contributions presented in this study are included in the article. Further inquiries can be directed to the corresponding author.

**Acknowledgments:** The authors acknowledge to: Comisión de Fomento a las Actividades Académicas, Estimulo al desempeño de los Investigadores; Sistema Nacional de Investigadores, Secretaria de Ciencia, Humanidades, Tecnología e Innovación (SECIHTI) México; and the Instituto Politécnico Nacional—SIP for 2389 multidisciplinary project (20251162, 20253472 and 20250704).

**Conflicts of Interest:** The authors declare no conflicts of interest.

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
