# Peer review of "Facile Reversible Eu2+/Eu3+ Redox in Y2SiO5 via Spark Plasma Sintering: Dwell Time-Dependent Luminescence Tuning"

_inorganics, doi:10.3390/inorganics13100325_

Round 1

Reviewer 1 Report

Comments and Suggestions for Authors

The manuscript submitted by Fernando Juárez-López reports the Eu²⁺/Eu³⁺ redox in Y₂SiO₅ via spark plasma sintering, and the influence of the dwell time on luminescence, the manuscript is overall not well prepared, and major revision is needed, some comments are as follows:

1 Why doping concentration of 1 and 2.5 at.% were chosen, and why only two concentrations were prepared. There is basically no difference in the luminescence properties for the two samples in the following analysis.

2 The crystal structure of X1 and X2 should be given in figures with polyhedrons for cation provided and described in detail.

3 The evidence for the formation of Eu2+ is insufficient. XPS analysis should be provided.

4 The analysis of figure 2 seems unrelated with the content of work.

5 The figures are not well prepared.

6 Photographs of the samples under both natural light and UV excitation should be provided.

7 Format mistakes can be found.

Author Response

Reviewer 1

The authors would like to thank the reviewer for his time and dedication in improving our manuscript. We hope we have answered the reviewer's questions satisfactorily. We would also like to inform you that a professional English editing service has proofread our manuscript. We attach the evidence. Thank you.

The manuscript submitted by Fernando Juárez-López reports the Eu²⁺/Eu³⁺ redox in Y₂SiO₅ via spark plasma sintering, and the influence of the dwell time on luminescence, the manuscript is overall not well prepared, and major revision is needed, some comments are as follows:

1 Why doping concentration of 1 and 2.5 at.% were chosen, and why only two concentrations were prepared. There is basically no difference in the luminescence properties for the two samples in the following analysis.

We appreciate the reviewer's observation, since the reason for choosing the Eu concentration was not mentioned. It is known that luminescent properties are a function of the content, and that as the concentration increases, the intensity increases. However, this effect has a limit, since after a certain concentration, the luminescent intensity decreases drastically due to the "quenching" effect that occurs when two Eu cations are too close to each other, such that the energy, instead of "leaving" the system, remains resonating between both cations. Although the content at which this phenomenon occurs can vary, usually for ceramic systems the maximum point is between 1 and 3%, which is why most studies carry out studies at 2.5%. In fact, specifically for Y2SiO5, Yogita Parganiha and coworkers (https://doi.org/10.1016/j.spmi.2014.11.010) analyzed the effect of Eu concentration and found that the maximum lies within the predicted range. References and an explanation of the choice of Eu content have been added to the text.

Regarding the differences in the results of both Eu contents, the main idea of ​​doing it was to be able to compare if there is a difference in the kinetic velocity at which Eu can be reduced in the SPS system, and indeed it was shown that there is a difference, and we would like to highlight that: During the SPS process, as seen in figure 5, the reduction is carried out at a higher velocity for 2.5% Eu, which makes sense since with a higher reactant content, kinetically the velocity will increase. This directly impacts the luminescent properties, as evidenced in the CIE diagrams, where, on the other hand, having a higher Eu content in the 2.5% sample, although the velocity is higher, more time is also required to increase the reduction efficiency, so it is possible to obtain white light after 15 min of process. We believe that these results highlight the importance of modulating both the Eu content and the time in the SPS. Corresponding paragraphs have been added to the sections to highlight these results.

2 The crystal structure of X1 and X2 should be given in figures with polyhedrons for cation provided and described in detail.

Figures have been added in the XRD results for better understanding, while the description has been added in the introduction.

3 The evidence for the formation of Eu2+ is insufficient. XPS analysis should be provided.

We understand the reviewer's point of view; however, unfortunately, we do not have access to this technique at the moment. However, we would like to mention that the luminescence results fully demonstrate the presence of the Eu2+ cation, since the emission observed at 365 nm is exclusive to this cation. In fact, as can be seen in the pre-SPS luminescence results, this emission is absent. In fact, luminescence studies are used to demonstrate the presence of specific oxidation states of rare-earth cations, since each state exhibits a characteristic and specific emission. We hope this observation will be sufficient for the reviewer. Thank you very much.

4 The analysis of figure 2 seems unrelated with the content of work.

The results in Figure 2 have been modified, adding further analysis of the results. Additionally, the following text has been added initially: "In order to verify the correct sintering of the Y2SiO5 powders during the SPS process, the shrinkage of the samples was analyzed during the process, in order to establish whether the powders were effectively compacted”. This is in order to introduce the reader that before analyzing the luminescent properties of the system, it is necessary to determine if the compaction process actually existed.

5 The figures are not well prepared.

We greatly appreciate the comment, the figures have been modified for better understanding.

6 Photographs of the samples under both natural light and UV excitation should be provided.

Figure 7e has been added, showing the coupon under natural light, at 254 nm and 365 nm.

7 Format mistakes can be found.

Formatting errors have been corrected. Thank you.

Reviewer 2 Report

Comments and Suggestions for Authors

This study focuses on Y₂SiO₅ powder doped with Eu³⁺ prepared by the sol-gel method and treated with spark plasma sintering (SPS) technology. It was found that the SPS process leads to the partial reduction of Eu³⁺ to Eu²⁺, and this reduction phenomenon is closely related to the holding time (HT) in the SPS chamber. The results indicate that the SPS sintering method can effectively regulate the valence state of luminescent centers, thereby enabling control over optical properties.

  1. Several key images (e.g., Figures 1and 6) suffer from small labels or missing legends, making interpretation difficult. Improving figure readability would significantly enhance clarity.
  2. There may be parts of the formula that are not shown, so please correct them.
  3. The content of the article is less innovative.
  4. Key mechanisms lack validation. Section 2.3.3 proposes that the reduction of Eu³⁺ is caused by the synergistic effect of CO and oxygen vacancies but fails to provide direct evidence (e.g., EPR detection of oxygen vacancies, atmospheric experiments to rule out interference).
  5. Comparative experiments are missing. No data from traditional sintering methods (e.g., muffle furnace) are provided to demonstrate the uniqueness of SPS.
  6. The conclusion emphasizes "optical encryption applications" but does not include demonstration experiments (e.g., multi-color encoded patterns).
Comments on the Quality of English Language

Need to refine language expression.

Author Response

The authors would like to thank the reviewer for his time and dedication in improving our manuscript. We hope we have answered the reviewer's questions satisfactorily. We would also like to inform you that a professional English editing service has proofread our manuscript. We attach the evidence. Thank you.

Reviewer 2

This study focuses on Y₂SiO₅ powder doped with Eu³⁺ prepared by the sol-gel method and treated with spark plasma sintering (SPS) technology. It was found that the SPS process leads to the partial reduction of Eu³⁺ to Eu²⁺, and this reduction phenomenon is closely related to the holding time (HT) in the SPS chamber. The results indicate that the SPS sintering method can effectively regulate the valence state of luminescent centers, thereby enabling control over optical properties.

  1. Several key images (e.g., Figures 1and 6) suffer from small labels or missing legends, making interpretation difficult. Improving figure readability would significantly enhance clarity.

The figures have been corrected so that all data within them is visible. They have also been resized to the same size.

  1. There may be parts of the formula that are not shown, so please correct them.

The authors appreciate the reviewer's time in correcting our manuscript. However, in this case, we were unable to identify the error the reviewer is referring to. We believe the confusion probably stems from the sign  , which is a specific nomenclature that represents an oxygen vacancy. If it is another error, we would appreciate it if the reviewer could indicate which line we should correct it. Thank you very much.

  1. The content of the article is less innovative.

We appreciate the reviewer's comment, and would like to mention that the authors believe that there are several key points of our work that are interesting: (1) Although the SPS process is known to create the formation of oxygen vacancies and free CO when using a graphite die, the present work is the first to explore this property to reduce the oxidation state of rare earth cations in a controlled manner; (2) It was shown that this process not only leads to the reduction of the luminescent cation, but also does so in different proportions depending on the initial cation content and the sintering time, which opens the possibility to appropriately modulate the luminescent response of the material; (3) It was shown that the process is reversible, and the cation can be reoxidized with a simple thermal treatment; and (4) A new process is proposed for the production of luminescent phosphors where in a single step it is possible to both modulate the ratio of two cations (where one is reduced) and at the same time sinter the material. We hope that the authors have been able to express these points with the corrections made to the manuscript.

  1. Key mechanisms lack validation. Section 2.3.3 proposes that the reduction of Eu³⁺ is caused by the synergistic effect of CO and oxygen vacancies but fails to provide direct evidence (e.g., EPR detection of oxygen vacancies, atmospheric experiments to rule out interference).

The authors appreciate the feedback and understand the author's concerns; however, we regret not having access to the EPR technique. However, we would like to mention that the hypothesis of this work is based on the possibility of cation reduction due to the probable combined mechanism of vacancy formation and CO formation. Regarding the first point, there is extensive literature that supports that vacancies occur due to rapid heating and the Jouel effect within the SPS (see, for example, https://doi.org/10.1002/cctc.201901549; https://doi.org/10.1007/s10832-021-00273-4, or https://doi.org/10.1016/j.ceramint.2024.04.017). On the other hand, the formation of CO has also been extensively reported (e.g. https://doi.org/10.1016/j.actamat.2014.10.030 or DOI: 10.1111/jace.12657). In the presented case, the luminescent results clearly demonstrate that the reduction of the Catio Eu3+ to Eu2+ has taken place, since in all the samples the emission of the blue band of the second is clear, which does not exist for the first. Therefore, we believe that taking advantage of these two SPS reduction processes has proven to be possible. The authors understand that further experiments will be necessary (and we are working on them for future investigations), but the objective of this manuscript is to demonstrate that tuning luminescent properties using this technique is possible.

  1. Comparative experiments are missing. No data from traditional sintering methods (e.g., muffle furnace) are provided to demonstrate the uniqueness of SPS.

The authors understand the reviewer's opinion, but the literature has amply demonstrated that the SPSP method is superior to traditional sintering methods in that it is faster and allows for greater densification. Likewise, the objective of our study is to specifically exploit the formation of vacancies and CO, all of which can be used for luminescent properties. Therefore, the use of traditional sintering methods cannot be compared.

  1. The conclusion emphasizes "optical encryption applications" but does not include demonstration experiments (e.g., multi-color encoded patterns).

The reviewer is absolutely right, and the aforementioned phrases from the introduction have been removed. Thank you.

Reviewer 3 Report

Comments and Suggestions for Authors

The manuscript under review, entitled “Facile Reversible Eu²⁺/Eu³⁺ Redox in Y₂SiO₅ via Spark Plasma Sintering: Dwell Time-Dependent Luminescence Tuning”, deals with the synthesis and characterization of luminescent functional materials with clear practical value. The aim and the results of the study fall into the scope of the Journal, namely the 1st and 3rd topics of the Scope list. The results are sound and discussed by the authors, but there are several minor corrections needed to improve the quality of the manuscript.

1) Please include a brief description of the measurements of shrinkage under sintering in the manuscript.

2) It is better to remove the emission spectrum obtained at 365 nm from Fig. 3b as its low intensity may confuse the reader. It is hard to see from the excitation spectrum that 365 nm resides in the gap between bands peaked at 362 and 380 nm.

3) It is worth marking the y-axis in Fig. 4 as “Normalized intensity” as it is likely the spectra were normalized.

4) Lines 275-276. Please clarify the sentence “the electron affinity, two possibilities exist: one for the Y–SiO₄ bond with ea=3.89ea= 3.89ea=3.89 eV, and another for the Y–O bond with ea=1.89ea = 1.89ea=1.89 eV ”. Also, check the ref 68, as there is nothing in it about electron affinity in Y-SiOx systems.

5) What the TT is stated for in Table 5. Please specify the abbreviation.

Comments on the Quality of English Language

The English should be improved. There are some points below.

1) While SPS states for Spark Plasma Sintering, “SPS sintering method” should be reduced to “SPS method” or use the full name of the technique.

2) It is not clear what a coupon is. After reading the manuscript, it is likely that “coupon” is stated for “disc”. Please clarify.

3) Fig. 1 capture. It is said, “Structural evolution of Y2SiO5:Eu3+.(a) Sol-gel derived powders, T= 1000°C, (b) SPS coupons of 1.0 at.%. Eu as a function of HT. c) SPS coupons of 2.5 at.%. Eu as a function of HT. ”

I recommend using “XRD patterns of Y2SiO5:Eu3+: (a) sol-gel derived powders, annealed at T= 1000°C, (b) SPS treated samples doped with 1.0 at.%. Eu at different HT. c) SPS treated samples doped with 2.5 at.%. Eu at different HT”. The authors show not the samples but XRD, so it should be depicted in the caption.

4) instead of “intramolecular ⁷F → ⁵D transitions ” it is better to say “intracenter ⁷F → ⁵D transitions ”

5) Starting from Section 2.3.2 the authors used section headings as “Sintered Y₂SiO₅: Eu³⁺ and SPS. ….” It is better to reduce it to “Sintered Y₂SiO₅: Eu³⁺”

Author Response

The authors would like to thank the reviewer for his time and dedication in improving our manuscript. We hope we have answered the reviewer's questions satisfactorily. We would also like to inform you that a professional English editing service has proofread our manuscript. We attach the evidence. Thank you.

Reviewer 3

The manuscript under review, entitled “Facile Reversible Eu²⁺/Eu³⁺ Redox in Y₂SiO₅ via Spark Plasma Sintering: Dwell Time-Dependent Luminescence Tuning”, deals with the synthesis and characterization of luminescent functional materials with clear practical value. The aim and the results of the study fall into the scope of the Journal, namely the 1st and 3rd topics of the Scope list. The results are sound and discussed by the authors, but there are several minor corrections needed to improve the quality of the manuscript.

  • Please include a brief description of the measurements of shrinkage under sintering in the manuscript.

The authors appreciate the observation, and in the corresponding part of materials and methods the following sentence has been added: "On the other hand, to ensure the correct compaction process, shrinkage is monitored as a function of both temperature and time. Shrinkage can be defined as: dL/L0, where L0 is the initial thickness of the specimen, and is therefore considered to be the instantaneous displacement rate."

  • It is better to remove the emission spectrum obtained at 365 nm from Fig. 3b as its low intensity may confuse the reader. It is hard to see from the excitation spectrum that 365 nm resides in the gap between bands peaked at 362 and 380 nm.

The authors appreciate the comments and, as suggested by the author, the 365 nm spectrum of the powders has been removed from the figure. Regarding the 365 nm excitation, the spectrum shows that the excitation band is very wide and that the maximum can actually vary. However, the authors wish to point out that the 365 nm excitation was chosen due to the existence of commercial LEDs at that wavelength, which falls within the maximum excitation zone.

  • It is worth marking the y-axis in Fig. 4 as “Normalized intensity” as it is likely the spectra were normalized.

Thanks for the comment and the figure has been corrected as suggested by the reviewer.

  • Lines 275-276. Please clarify the sentence “the electron affinity, two possibilities exist: one for the Y–SiO₄ bond with ea=3.89ea= 3.89ea=3.89 eV, and another for the Y–O bond with ea=1.89ea = 1.89ea=1.89 eV ”. Also, check the ref 68, as there is nothing in it about electron affinity in Y-SiOx systems.

We deeply appreciate the comment from the receiver, as the wording was indeed incorrect, and even more so, the reference was incorrect. The reference to the electron affinity of SiO4 was added, considering SiO2 as the reference, since there are no specific measurements for silicates. Furthermore, the reference to the electron affinity corresponding to O2 was corrected. Thank you.

5) What the TT is stated for in Table 5. Please specify the abbreviation.

The aforementioned abbreviation referred to samples that, after being processed at the SPS, were oxidized again in air at 1000°C. However, to avoid confusion, we have restored the abbreviation and added the legend "Oxidized" for clarity. Thank you.

Round 2

Reviewer 1 Report

Comments and Suggestions for Authors

The authors improved the manuscript and the manuscript is acceptable for publication.